# Robust ML Auditing using Prior Knowledge

Jade Garcia Bourrée [* 1 2 3]   Augustin Godinot [* 1 2 3 4]   Sayan Biswas [5]   Anne-Marie Kermarrec [5]
Erwan Le Merrer [2]   Gilles Tredan [6]   Martijn de Vos [5]   Milos Vujasinovic [5]

## Abstract

Among the many technical challenges to enforcing AI regulations, one crucial yet underexplored problem is the risk of audit manipulation. This manipulation occurs when a platform deliberately alters its answers to a regulator to pass an audit without modifying its answers to other users. In this paper, we introduce a novel approach to manipulation-proof auditing by taking into account the auditor's prior knowledge of the task solved by the platform. We first demonstrate that regulators must not rely on public priors (*e.g.*, a public dataset), as platforms could easily fool the auditor in such cases. We then formally establish the conditions under which an auditor can prevent audit manipulations using prior knowledge about the ground truth. Finally, our experiments with two standard datasets illustrate the maximum level of unfairness a platform can hide before being detected as malicious. Our formalization and generalization of manipulation-proof auditing with a prior opens up new research directions for more robust fairness audits.

## 1. Introduction

Machine learning (ML) models are becoming central to numerous businesses, industrial processes, and administrations. Such models are being employed in high-stakes domains where ML-driven decisions can have profound impacts on individuals and communities (Rudin, 2019).

For instance, financial institutions have been leveraging ML-driven systems to evaluate loan applications based on attributes like income, credit score, and employment history

(West, 2000). Given the far-reaching consequences of these applications, ensuring the fairness (Mehrabi et al., 2021) and regulatory compliance (Petersen et al., 2022) of such models is paramount.

Independent *fairness audits* serve as a critical tool for assessing the fairness of ML models and ensure that model providers remain accountable to the public (Birhane et al., 2024; Raji, 2024; Raji et al., 2022). As models are placed in production, auditors rely on black-box interactions, where queries are sent to the model, and the responses are analyzed to identify potential fairness violations (*e.g.*, see (Kim et al., 2019)). However, this reliance on black-box audits leaves the process vulnerable to *manipulations* by the platform, also known as *fairwashing*. Regulatory practices currently require auditors to notify platforms in advance of an audit. Platforms can thus strategically alter the model or its responses during the audit to create the appearance of fairness, effectively concealing underlying biases and unfair practices from the auditor while maintaining operational efficiency for its users. Consider, for example, a social media platform that employs an ML model to moderate content, automatically removing posts deemed harmful or misleading. During a fairness audit, the platform could deploy a more lenient moderation model that appears unbiased, only to revert to a stricter, potentially biased version once the audit concludes, effectively concealing unfair treatment of certain user groups.

In fact, such discrepancies between a platform's behavior during audits and its real-world operations have been observed. Initially created as part of the Social Science One project, a data sharing program by Meta encountered a major setback when consistency issues were discovered in the data provided to scientists (Timberg, 2021). Similarly, a collaboration between Meta and independent scientists, studying the polarization effects of Facebook's recommendation algorithm, recently faced criticism over discrepancies found between the algorithm's behavior before and during the audit (Ribeiro, 2024). Academic studies show that fairness audits are easily manipulatable, whether the platform is required to prove its fairness through the release of a public dataset (Fukuchi et al., 2020), through the explanation of decisions (Aïvodji et al., 2021; Shamsabadi et al., 2022; Le Merrer & Trédan, 2020; Aïvodji et al., 2019), or

---

[*]Equal contribution  [1]Université de Rennes, Rennes, France [2]Inria, Rennes, France [3]IRISA/CNRS, Rennes, France [4]PEReN, Paris, France [5]EPFL, Lausanne, Switzerland [6]LAAS, CNRS, Toulouse, France. Correspondence to: Augustin Godinot <augustin.godinot@inria.fr>, Jade Garcia Bourrée <jade.garcia-bourree@inria.fr>.

*Proceedings of the 42nd International Conference on Machine Learning*, Vancouver, Canada. PMLR 267, 2025. Copyright 2025 by the author(s).

through black-box query interactions (Yan & Zhang, 2022; Garcia Bourrée et al., 2023; Godinot et al., 2024). The potential for manipulation underscores the need for more robust auditing strategies.

This work presents a novel theoretical framework and a practical implementation for preventing manipulations by the platform. Our analysis starts from a simple observation: auditors can readily collect labeled data, reflecting the platform's service from independent sources – a common practice whose theoretical and empirical implications remain unexplored. For example, in the moderation example discussed earlier, the auditor could have some undeniable evidence at hand, to confront the model under scrutiny, *e.g.*, "A post with this content *must* pass the moderation filter, otherwise there is some bias on a protected feature of the user profile". Thus, by incorporating this dataset, the auditor can independently verify the platform's responses, cross-referencing them against known ground truth labels. By combining black-box interactions with prior knowledge from the labeled dataset, our method enables more reliable detection of fairness violations while reducing the reliance on assumptions about the platform's behavior. Specifically, we aim to answer the following research question: *Can the auditor's prior knowledge of the ground truth prevent fairwashing in fairness audits?*

Our paper makes the following three contributions:

- We introduce and analyze a new fairness auditing approach for black-box interactions where the auditor has access to prior knowledge about the platform and the ML task (Section 3).

- We theoretically analyze how much unfairness a platform can conceal given the auditor's prior knowledge. For any auditor priors, our results highlight the importance of keeping the auditor's prior knowledge private (Section 3). For the dataset prior we introduce, we establish bounds on the concealable unfairness when the auditor prior remains confidential (Section 4).

- By simulating fairness audits on multiple tabular and vision datasets, we provide a more nuanced understanding of how our framework should be implemented. Our experiments offer insights into setting the detection threshold used to identify manipulations (Section 5).

## 2. Background: Auditing ML Models

This work studies fairness audits of ML decision-making systems under manipulation by the model-hosting platform. We first formalize the decision-making system and then introduce the dynamics of fairness auditing.

**ML decision-making systems**  From feature transforms to specific business rules, modern ML decision-making systems can be remarkably complex. We abstract all this complexity by modeling the entire system as a function $h : \mathcal{X} \to \mathcal{Y}$ (*e.g.*, $h$ can be a ML model). The set of possible queries $\mathcal{X}$ is called the *input space*, and the set of possible answers is called the *output space*. We consider binary classification problems, which is in line with related work in the domain of ML fairness analysis (Yan & Zhang, 2022; Godinot et al., 2024). Each query is associated with a protected attribute $a \in \mathcal{A}$, which the platform is legally required not to discriminate against. Examples of such attributes include gender, age, or race and are typically defined by law. The platform has access to the protected attribute $a$ either as a feature of the input space $\mathcal{X}$ or by a proxy (*e.g.*, looking at the name of the person to determine the gender). We define $\mathcal{D}$ as the data distribution on $\mathcal{X} \times \mathcal{A}$. For any subset $S \subset \mathcal{X} \times \mathcal{A}$ and protected feature value $a \in A$, we will write $S_a = \{x \mid (x, a') \in S, a' = a\}$ and $h(S) = \{h(x) \mid (x, a) \in S\}$. Throughout the paper, when it is clear from the context we will abuse the $S$ notation: $S$ will either be a subset of $\mathcal{X}$, $\mathcal{X} \times \mathcal{A}$ or $\mathcal{X} \times \mathcal{A} \times \mathcal{Y}$.

This work analyses how the platform can manipulate its model to pass a fairness audit, we now define relevant notation for this. The space of models that the platform can implement is called the *hypothesis space* $\mathcal{H}$. The loss function $L : \mathcal{H} \times (\mathcal{X} \times \mathcal{Y}) \to \mathbb{R}$ measures the discrepancy between predictions and ground truth values. For a given hypothesis $h \in \mathcal{H}$, its expected loss over the distribution $\mathcal{D}$ is $L(h, \mathcal{D}) = \mathbb{E}_{(x,a) \sim \mathcal{D}} [\ell(h(x), x, a)]$ with $\ell$ is a loss function that quantifies the error of $h(x)$ given a single input $x$ and its protected attribute $a$.

**ML auditing**  An ML audit is "any independent *assessment* of an identified *audit target* via an *evaluation* of articulated expectations with the implicit or explicit objective of *accountability*" (Birhane et al., 2024). A ML audit involves three entities. The *platform* is the entity hosting the ML decision-making system. The *users* are those using the service hosted by the platform. The *auditor* is the entity conducting the audit to verify whether the ML model is compliant for *all* users. The auditor could be a state regulator, a consulting firm, or even a group of users.

**Fairness metric**  In this work, we consider ML audits targeting the *fairness* of the studied system. Specifically, the auditor chooses a fairness metric and sends queries to the platform to determine whether the platform abides by their fairness criterion. Among all the (un-)fairness metrics, we study *Demographic Parity (DP)* (Calders et al., 2009), which is commonly used in the fairness evaluation literature thanks to its simplicity. DP is defined as follows:

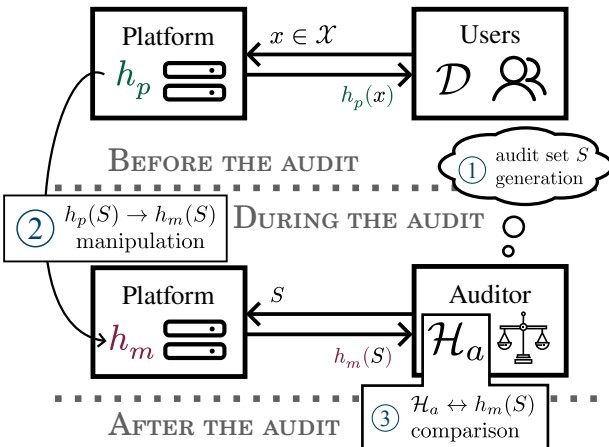

Figure 1. The auditing process as conducted by an auditor, which proceeds in three steps. The platform exposes a model $h_p$ to the users. To appear fair to the auditor while not deteriorating the utility for its users, the platform manipulates its answers on the audit set $S$.

$$\mu(h) = \begin{aligned} &\mathbb{P}_{(X,A)\sim\mathcal{D}}\left(h(X) = 1 | A = 1\right) \\ &- \mathbb{P}_{(X,A)\sim\mathcal{D}}\left(h(X) = 1 | A = 0\right) \end{aligned} \quad (1)$$

For a platform, DP is the easiest metric to manipulate (Yan & Zhang, 2022; Ajarra et al., 2024) as it only depends on the *outcome* of the ML model and not on its performance on the different protected groups. Thus, a platform can artificially adjust outputs, *e.g.*, providing more positive outcomes for an underrepresented group. To decide whether a platform passes the audit or not, the auditor builds an *audit set* $S \subset \mathcal{X} \times \mathcal{A}$ and evaluates the plug-in DP estimator: $\hat{\mu}(h, S) = \frac{1}{|S_1|}\sum_{x \in S_1} \mathbb{1}\{h(x) = 1\} - \frac{1}{|S_0|}\sum_{x \in S_0} \mathbb{1}\{h(x) = 1\}$. Based on $\mu(h)$, we also define the set of fair models $\mathcal{F} = \{h \in \mathcal{Y}^{\mathcal{X}} : \mu(h) = 0\}$.

## 3. Enhancing Black-box Auditing with a Prior

Since a malicious platform can manipulate the DP metric with relative ease, the auditor has to find ways to prevent these manipulations (*e.g.*, using a different metric) or to detect them. In this section, we explore the latter. To detect manipulations, the auditor must use *prior* knowledge about what constitutes a "likely set of answers" on its audit dataset $S$. Then, using this prior, they would be able to estimate the likelihood that the received set of answers $h_m(S)$ has been manipulated.

### 3.1. Modeling the Auditor Prior

Previous work has demonstrated that prior knowledge is both a practical and an essential tool for auditing, yet the notion of an auditor prior has not been explicitly leveraged

in the analysis of fairness audits. We define an *auditor prior* as follows.

**Definition 3.1** (Auditor prior). The *auditor prior* is a set of models $\mathcal{H}_a \subset \mathcal{Y}^{\mathcal{X}}$ that the auditor can reasonably expect to observe given her knowledge of the decision task by the platform.

For example, in (Tan et al., 2018), the authors study feature importance by training two models — one on a public dataset and another via distillation of the audited ML model — and comparing the resulting models. Using a more theoretical approach, Yan & Zhang and Godinot et al. explored the case of an auditor knowing the hypothesis class of the platform, *i.e.*, $\mathcal{H}_a = \mathcal{H}$. Ajarra et al. proposed to use an assumption about the Boolean Fourier coefficients of $\mathcal{H}$ to construct $\mathcal{H}_a$. Finally, Garcia Bourrée et al. and Shamsabadi et al. used side-channel access (*e.g.*, an additional API or explanations) to the ML model to define $\mathcal{H}_a$ and derive guarantees on the measured fairness. In Section 4, we introduce a labeled dataset $D_a$ that the auditor will leverage to define $\mathcal{H}_a$. Definition 3.1 captures all of the situations above and allows to formulate general results about the problem of robust auditing.

**The auditing process** The auditing process consists of three steps which we visualize in Figure 1. Here, $h_p$ refers to the model that the platform exposes to its users (the top part of Figure 1) and $h_m$ refers to the model exposed to the auditor (bottom part of Figure 1). First, the auditor builds an audit set $S \subset \mathcal{X}$ and sends the queries in $S$ to the platform (step ①). The platform receives $S$ *all at once* and computes the answers using its model $h_p$. To appear fair if it is not, the platform projects its labels $h_p(S)$ on the set $\mathcal{F}$ of fair models. This defines a manipulated model $h_m$ and the answers $h_m(S)$ the platform will send to the auditor (step ②). The auditor receives $h_m(S)$ and exploits these samples to evaluate whether the platform is fair ($h_m \in \mathcal{F}$) and honest ($h_p = h_m$), (step ③). Since the auditor does not have direct access to $h_p$, they compare $h_m$ to their prior $\mathcal{H}_a$ to decide whether the platform is honest or malicious. Thus, the auditor tests the two following properties of $h_m$:

$$\text{Is the platform fair?} \quad h_m \overset{?}{\in} \mathcal{F} \quad (2)$$

$$\text{Is the platform honest?} \quad h_m \overset{?}{\in} \mathcal{H}_a \quad (3)$$

For dataset priors (*i.e.*, when $\mathcal{H}_a$ is a ball, see Section 4), we draw $\mathcal{F}$ and $\mathcal{H}_a$ in Figure 2. Given a model $h_m$, the fairness audit is equivalent to checking if $h_m$ belongs to the blue shaded area. In the example of Figure 2, the platform would be flagged as malicious as $h_m$ belongs to $\mathcal{F}$ but not to $\mathcal{H}_a$.

**Online v.s. batch auditing** Note that we assume that the platform receives all audit queries at once and that it is possible to detect all the audit queries. In practice, the queries

are usually issued online (that is, one-by-one) by the auditor, through web-scraping or through an API. Compared to online auditing, it is easier for the platform to manipulate an audit if it knows all the audit queries before having to answer. On the other hand, because the auditor has to send all their queries at once, they cannot use the answers of the platform to actively guide the generation of the audit questions (*e.g.*, as in (Yan & Zhang, 2022; Godinot et al., 2024)). Ultimately, our setting is built as a worst-case analysis of the auditing game for the auditor.

**Auditing axioms**   To avoid trivial audits, we add two modeling assumptions. The first assumption ensures that the auditors' prior is correct so that a honest platform does not appear as lying. The second assumption asserts that an audit is necessary, otherwise the auditor could directly conclude from his prior that the platform is unfair. That is to say, the auditor should never flag a honest platform malicious. In particular, the auditor must have a prior that is close to the ground truth. Those assumptions are expressed as:

$$h_p \in \mathcal{H}_a \qquad \text{and} \qquad \mathcal{H}_a \cap \mathcal{F} \neq \emptyset. \qquad (4)$$

### 3.2. On Public Auditor Priors

A typical auditor proceeds in the following way. Upon examining a platform's model $h_m$, the auditor must first understand the task addressed by $h_m$ and what constitutes a "good-performing model" on this task. In our moderation example, the auditor might try to look for public moderation datasets to test the performance of $h_m$ using a few examples. It might also look for publicly-available moderation models to compare their resulting input/output pairs with those of $h_m$. Unfortunately, our first remark is that regardless of the prior the auditor might construct, if these models are public (or at least known by the platform), the platform will always be able to manipulate the audit:

**Theorem 3.2.** *Assume the platform knows $\mathcal{H}_a$, it can then always pick $h_m \in \{\mathcal{H}_a \cap \mathcal{F}\}$ to appear both fair and honest.*

*Proof.* First, recall that by definition the platform knows $\mathcal{F}$. Assume that the dataset prior is public, the platform also knows $\mathcal{H}_a$. Hence the platform can compute $\mathcal{F} \cap \mathcal{H}_a$. As by assumption, $\mathcal{F} \cap \mathcal{H}_a \neq \emptyset$ (Equation (4)), the platform can pick any model $h_m \in \mathcal{F} \cap \mathcal{H}_a$. □

In the case of (Shamsabadi et al., 2022), the platform perfectly knows $\mathcal{H}_a$ (because the $\mathcal{H}_a$ is coming from queries of its model) so the detector is subject to this manipulation (called *Irreducibility* in the paper). In Yan & Zhang's work, $\mathcal{H}_a$ is the hypothesis class $\mathcal{H}$ of the platform, communicated to the auditor before the audit. Theorem 3.2 provides a novel view on the impossibility results that were later proved in (Godinot et al., 2024).

## 4. Using Labeled Datasets for More Robust Audits Against Manipulations

In an ideal, yet unrealistic audit scenario, the auditor would have access to non-manipulated answers from the original platform model $h_p$. The prior $\mathcal{H}_a$ would then be the set of models that agree with these non-manipulated answers and would allow the auditor to detect inconsistencies between the original $h_p$ and manipulated $h_m$ models. Yet in general, the auditor does not have access to such non-manipulated answers.

As an alternative, we propose to study the use of a private (because of Theorem 3.2) dataset $D_a$, collected by the auditor to construct the auditor prior $\mathcal{H}_a$. This idea (coupled with an assumption on the hypothesis class) has been studied experimentally (Tan et al., 2018) but the more recent theoretical works on robust auditing diverged towards studying priors on the model itself rather than on the data (Shamsabadi et al., 2022; Yan & Zhang, 2022; Ajarra et al., 2024). In the following, we define what a dataset prior is, and study the guarantees an auditor can achieve using this prior. Unless noted otherwise, in this section and in Section 5, $\mathcal{H}_a$ will denote the dataset prior.

**Definition 4.1** (Dataset prior). Let $D_a = (X_a, A_a, Y_a) \in \mathcal{X}^n \times \mathcal{A}^n \times \mathcal{Y}^n$ be a labeled dataset the auditor has access to. The dataset prior $\mathcal{H}_a$ is defined as the set of models that have a reasonable risk on $D_a$.

$$\mathcal{H}_a = \left\{ h \in \mathcal{Y}^{\mathcal{X}} : L(h, D_a) < \tau \right\}. \qquad (5)$$

To test if the platform is honest, the auditor needs to verify whether $h_m \in \mathcal{H}_a$, *i.e.*, whether $L(h_m, D_a) < \tau$. The risk threshold $\tau$ thus plays a big role in the guarantees the auditor will be able to achieve. We discuss the impact of $\tau$ in Section 4.2 and guidelines to set its value in Section 4.3, but first, we need to discuss the definition of optimal manipulation in Section 4.1.

### 4.1. Optimal Manipulation

Given the audit set $S$ and its model $h_p$, the objective of a manipulative platform is to create a set of answers $h_m(S)$ that appear fair to the auditor but also do not raise suspicions. Ideally, the platform would like to know the auditor prior $\mathcal{H}_a$ (see Theorem 3.2), but in the general case it cannot because it is not public information. As a consequence, the platform cannot directly optimize its answers to be expectable *and* fair. However, it still has cards up its sleeve; it already trained a model $h_p$ on a dataset $D$ that is close to that of the auditor $D_a$.

Thus, instead of searching $h_m$ in $\mathcal{H}_a \cap \mathcal{F}$, the platform can assume that its true model $h_p$ is expectable – that is, $h_p \in \mathcal{H}_a$ – and try to find a fair model $h_m \in \mathcal{F}$ while flipping as few labels as possible from $h_p$. Therefore, the

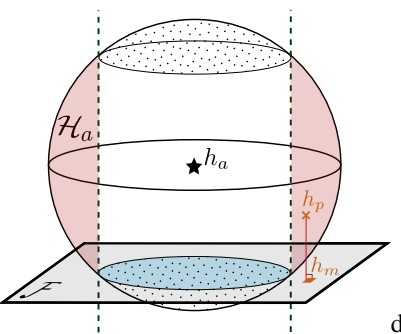

*Figure 2.* Representation of the auditor prior $\mathcal{H}_a$, the honest platform model $h_p$ and a corresponding malicious model $h_m$ on the fair $\mathcal{F}$ plane. The red area represents the area where platforms optimal manipulations are detected as dishonest: they fall outside of the blue region of $\mathcal{F}$

optimal manipulation is the projection of $h_p$ on $\mathcal{F}$:

$$h_m^* = \text{proj}_{\mathcal{F}}(h_p) = \underset{h \in \mathcal{F}}{\arg\min}\, d(h, h_p). \qquad (6)$$

The distance $d$ in Equation (6) is the value of risk $L$ of $h$ using the labels of $h_p$ as the ground truth. This scenario captures the fairwashing approach in (Aïvodji et al., 2021) in the context of explanation manipulations.

### 4.2. Achievable Guarantees

Following the second auditing axiom formulated in Equation (4), the original model of the platform is always expectable, *i.e.*, $h_p \in \mathcal{H}_a$. Thus, the manipulation detection test has no false positives, and the main quantity of interest to the auditor is the *manipulation detection rate*.

**Definition 4.2** (Detection rate). The probability $P_{uf}$ that the auditor correctly detects a manipulative platform with optimal manipulation is $P_{uf} = P(h_m^* \notin \mathcal{H}_a | h_p \in \mathcal{H}_a)$.

Estimating or computing $P_{uf}$ requires the knowledge of the distribution of models in $\mathcal{H}_a$. Unfortunately, unless they have access to the training pipeline of the platform, this model distribution is inaccessible to the auditor. To overcome this issue, we make the assumption of an *uninformative prior*: since the auditor does not know the model distribution in $\mathcal{H}_a$, they must assume it is uniform.

**Theorem 4.3** (Prior-Uniform detection rate). *Under the dataset prior of Definition 4.1 with $L$ defined as the $\ell_2$ norm, and the* uninformative prior *assumption, the probability that the auditor correctly detects a malicious platform trying to be fair is*

$$1 - \frac{1}{W_n}\left(\int_0^{\arccos(\delta/\tau)} \sin^n(\theta)d\theta - \frac{\delta}{\tau}\left(1 - \frac{\delta^2}{\tau^2}\right)^{(n-1)/2}\right).$$

*with $\delta = d(h_a, \mathcal{F})$, the distance of $h_a$ to $\mathcal{F}$ and $W_n$ is the $n$-term of Wallis' integrals.*

To gain intuition about the proof, we represent the audit case for $|S| = 3$ in Figure 2. By definition of the dataset prior, $\mathcal{H}_a$ is a ball of radius $\tau$, centered on $Y_a$, the labels given in the audit dataset $D_a$. The manipulation of a model $h_p$ can be detected only if the resulting model is outside of $\mathcal{H}_a$, as shown in orange on Figure 2. The probability of detection is thus $1$ minus the volume of original models $h_p$ whose projection on $\mathcal{F}$ lies outside on $\mathcal{H}_a$. This volume is highlighted in red in Figure 2. The detailed proof of Theorem 4.3 is deferred to Appendix A.

Theorem 4.3 highlights two key parameters to the auditor's success: the unfairness of the prior $\delta = d(h_a, \mathcal{F})$ and the expectability threshold $\tau$. If the dataset prior is perfectly fair (*i.e.*, $\delta = 0$), then the auditor has no chance to detect a manipulated model as non-expectable ($P_{uf} = 0$, Corollary A.5). On the other hand, Corollary A.4 proves that, if $\tau = \delta$ [1] then $P_{uf} = 1$. Finally, in Corollary 4.4, we derive a lower bound on $P_{uf}$ for the case $0 < \delta < \tau$. We provide the proof of Corollary 4.4 in Appendix A.

**Corollary 4.4** (Detection rate lower bound). *If $n$ is even,*

$$\frac{1}{W_n}\frac{\delta}{\tau}\left(1 - \frac{\delta^2}{\tau^2}\right)^{(n-1)/2} \leq P_{uf} \leq 1.$$

### 4.3. Practical Considerations and Discussion

In practice, $\tau$ is determined by the task difficulty, and the amount of data available to solve the task. One possibility to tune the value of $\tau$ is to use the error rate of current state-of-the-art models that solve the task at hand as a minimum value. We empirically explore this option in Section 5.4. An alternative, if the auditor has the resources, would be to train a set of models on the task and use them to calibrate $\tau$. We leave further exploration of the calibration of $\tau$ to future work.

On the other hand, the value of $\delta$ is determined by the audit set sampling procedure. In most cases, the audit set is sampled independently from a pre-specified audit distribution. In this case, the value of $\delta$ is fully determined. To regain some control over $\delta$, the auditor has to allow other audit set sampling strategies, at the expense of potential statistical bias in the fairness and accuracy estimations.

**Takeaway.** The auditor can always calculate *a priori* the probability to correctly detect a malicious platform trying to be fair. This probability depends on the ratio between unfairness $\delta$ of the auditor prior and the chosen risk threshold $\tau$, and depends on the audit budget $n = |S|$.

---

[1] Per our first axiom in Equation (4), we have that $\delta \leq \tau$.

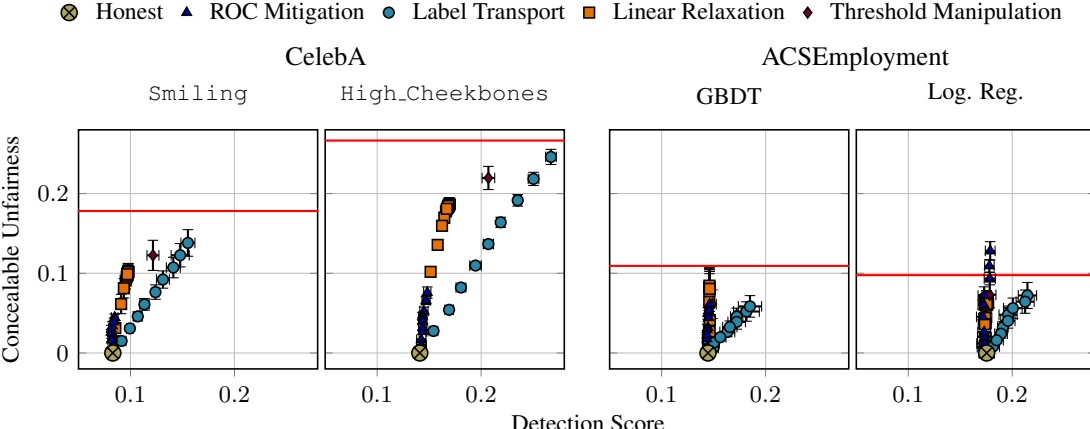

*Figure 3.* The concealable unfairness by the platform for different detection scores and manipulation strategies. We highlight this for two features of the CelebA dataset (left) and for two different ML models trained on the ACSEmployment dataset (right). The horizontal red line indicates the DP of the most unfair model without manipulation.

## 5. Empirical Evaluation

We now empirically quantify the extent to which the platform can manipulate the unfairness of its ML model. To that end, we study the *concealable unfairness*: the maximum level of unfairness a platform can hope to hide before being detected as malicious. First, we evaluate the effectiveness of different manipulation strategies and determine the optimal one. Since any practical fairness repair method can be used as a manipulation methods, we ask **(RQ1)** What is the best manipulation strategy implementation (Section 5.3)? Then, we study the dynamics of the concealable unfairness when the audit budget $|S|$ increases: **(RQ2)** Can the auditor always find an audit budget that prevents the platform from hiding any unfairness, *i.e.*, that always allows to flag the platform if malicious (Section 5.4)?

### 5.1. Experimental Setup

We conduct our experiments on tabular and vision modalities. The tabular dataset comes from the ACSEmployment task for the state of Minnesota in 2018, which is derived from US Census data and provided in folktables (Ding et al., 2021). The objective of this task is to predict whether an individual between the age of 16 and 90 is employed or not. As input features of the model $h_p$, we consider several attributes of the individual, including gender, race, and age. The fairness of the models is evaluated along the race attribute given in the dataset: one group consists of individuals identified as "white alone", while the other includes all remaining individuals.

For the vision modality, we study CelebA (Liu et al., 2015), which consists of images of celebrities along with several binary attributes associated with each image, such as whether the person in the photo is blond, smiling, or if the photo

is blurry. As input to a vision model, we use the image to predict one of the associated attributes. The target attribute varies across experiments and will be specified accordingly. Demographic Parity is evaluated along the gender attribute given in the dataset. For the ACSEmployment dataset, we train Gradient Boosted Decision Tree (GBDT) and Logistic Regression (Log. Reg.) models, while for CelebA, we train a LeNet convolutional neural network (Lecun et al., 1998). GBDT and Log. Reg. are trained using the default parameters of their respective implementations in SCIKIT-LEARN. Meanwhile, LeNet is trained irrespective of the target attribute using the Adam optimizer with a learning rate of $\gamma = 0.001$, a batch size of 32, and for two epochs, which is sufficient for the model to converge on all features. The code to run the experiments is available online.[2]

### 5.2. Implementing Optimal Audit Manipulations

In practice, computing the optimal manipulation $h_m = \text{proj}_{\mathcal{F}}(h_p)$ amounts to solving:

$$
\begin{aligned}
h_m(S) \ \in \ \arg\min \quad & L(h, \{(x, h_p(x)) : x \in S\}) \\
\text{s.t.} \quad & \hat{\mu}(h, S) < \tau
\end{aligned} \tag{7}
$$

We note that this problem is the same problem solved by in-processing and post-processing fairness repair methods (Caton & Haas, 2024). Thus, ironically, computing the optimal manipulation is equivalent to choosing the optimal fairness repair method. The only difference being on which set the fairness constraints and accuracy objectives are defined: the audit set $S$ instead of the training dataset. Thus, since any practical fairness repair method can be repurposed for manipulation, we adapted four classical fairness repair methods: ROC Mitigation (ROC) (Kamiran et al., 2012),

---

[2]See https://github.com/grodino/merlin.

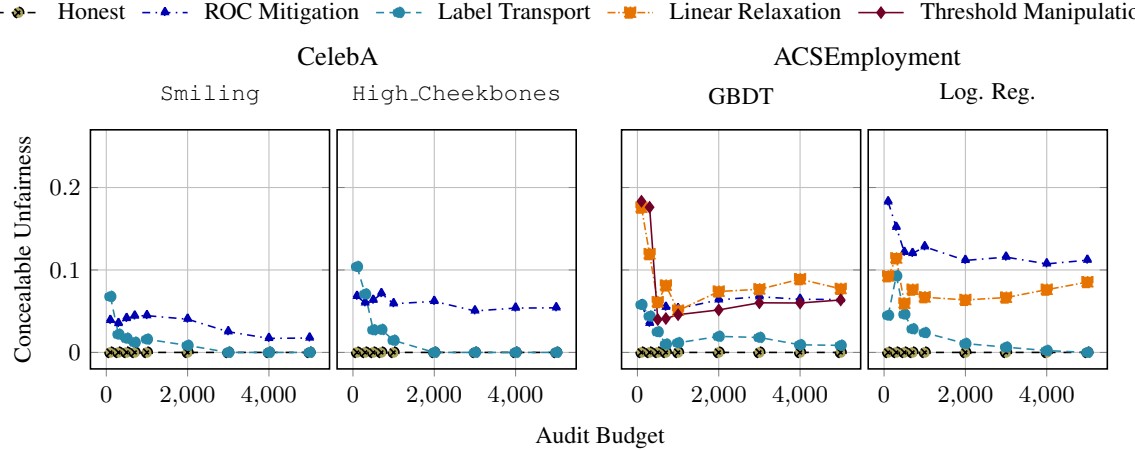

*Figure 4.* The concealable unfairness for different audit budgets (*i.e.*, data samples from the labeled dataset). We highlight this for two features of the CelebA dataset (left) and for two different ML models trained on the ACSEmployment dataset (right).

Optimal Label Transport (OT-L) (Jiang et al., 2019), Linear Relaxation (LinR) (Lohaus et al., 2020) and Threshold Manipulation (ThreshOpt) (Hardt et al., 2016).

## 5.3. Quantifying the Concealable Unfairness For Different Detection Scores

We next quantify the amount of fairness that a platform can hide for the different manipulation strategies above. This *concealable unfairness* $\Delta_\mu(h_p, h_m)$ is defined as the Demographic Parity gap between the manipulated and honest models.

$$\Delta_\mu(h_p, h_m) = |\hat{\mu}(h_m, S) - \hat{\mu}(h_p, S)| \quad (8)$$

To decide whether the model observed during the audit is manipulated, the auditor has to decide whether $h_m \in \mathcal{H}_a$ or not. To do so, the auditor estimates $L(h, D_a)$ by computing the *detection score* $\text{Detect}(h_m, S)$.

$$\text{Detect}(h_m, S) = \sum_{(x,y) \in S} \mathbb{1}\{h_m(x) \neq y\} \quad (9)$$

To build $(h_p, h_m)$ model pairs, we consider manipulation methods among ROC, OT-L, LinR and ThreshOpt, varying hyperparameter values when applicable. In Figure 3, we plot the value of the concealable unfairness $\Delta_\mu(h_p, h_m)$ against the detection score $\text{Detect}(h_m, S)$ computed by the auditor. We show the results of LeNet models trained on two CelebA targets (first and second subplots), and GBDT and Log. Reg. models trained on ACSEmployment (third and fourth subplots). The horizontal red lines indicates the DP of the most unfair model without manipulation.

First, we observe that for all the datasets, the platform can conceal significant amounts of unfairness: from 10 to 20

points differences between the two protected groups. Comparing the concealable unfairness values with the DP of the most unfair honest model (red horizontal line), we observe that the manipulation strategies almost all able to totally conceal the original model unfairness. Then, focusing on the x axis, the difference in $\text{Detect}(h_m, S)$ between the different honest models highlights the impact the performance of the platform's model should have on the detection threshold $\tau$. In fact, depending on the dataset and on the model, $\text{Detect}(honest, S)$ varies from $\sim 0.1$ to $\sim 0.2$. In Section 5.4, we explore a solution to setup the threshold.

## 5.4. Dynamics of the Concealable Unfairness as The Audit Budget Increases

The probability of detecting manipulations (via the the detection score) should intuitively increase as the auditor gains access to a larger number of data samples (*i.e.*, has a higher audit budget) since this allows for a more accurate comparison of $h_m$ with the data prior $\mathcal{H}_a$. In this experiment, we explore how well this intuition holds in practice. For this purpose, we fix the hyperparameters for each manipulation method by selecting those that result in the highest concealable unfairness for a given base model, as discussed in Section 5.3. Then, for each base model–target attribute pair, we determine the maximum concealable unfairness that a platform can achieve while ensuring that its detection score (see eq. 9) remains below the detection threshold. As proposed in Section 4 the threshold for each model is set to $1 - x$, where $x$ represents the maximum accuracy achieved when training a set of models on the corresponding target. This process is repeated for audit budgets ranging from 100 to 5,000.

The results of this experiment are shown in Figure 4. The two plots on the left display the results for CelebA using

the same base model but different target attributes, while the two plots on the right show results for ACSEmployment using the same target attribute but different base models. These results reveal two distinct cases. In the first case (CelebA `Smiling` in Figure 4), the concealable unfairness converges to zero as the audit budget increases. This is due to the low aleatoric uncertainty associated to the `Smiling` target. Since the task is easier, the accuracy range of models trained on `Smiling` is narrower, leading to a tighter detection threshold $\tau$. In the second case (all the other facets of Figure 4), the concealable unfairness remains nonzero despite an increasing budget.

Furthermore, in many cases, even with a high audit budget, some increase of unfairness remains undetectable by the auditor. Consequently, the platform retains some capacity to conceal unfairness even at high audit budgets. This stresses the hardness of the auditor's task in some configurations, and lead to a negative answer to **(RQ2)**. In that light, we also observe that –in response to **(RQ1)**– the Linear Relaxation and ROC Mitigation manipulation strategies are the most effective for a manipulative platform.

# 6. Related Work

**Fairwashing and rationalization** Addressing fairness issues often requires compromising model performance for advantaged groups which can discourage companies from embracing fair training practices (Zietlow et al., 2022; Zhao & Gordon, 2022). Companies have two incentives to pay attention to the impact of their system on society. The first incentive comes from regulatory efforts such as the Algorithmic Accountability Act (AAA) (Congress, 2022) (US) and the Digital Markets Act (DMA) (Union, 2022) (EU) that impose fairness, transparency, and accountability constraints on large digital platforms. Yet, how to enforce these regulations is still an open problem (Crémer et al., 2023). The second incentive is public image. Since fairness, transparency and accountability are laudable goal, audits, investigative journalism and certifications (Costanza-Chock et al., 2022) should force companies to pay attention to these objectives. However, both incentives are external: the platform just has to *appear* fair, transparent and accountable. This rationalization risk has been studied in the context of explanations fairwashing (Aïvodji et al., 2019; 2021; Shamsabadi et al., 2022).

**Fairness auditing** Fairness auditing evaluates ML models to ensure fairness and accountability, often without access to proprietary model internals (Ng, 2021). This black-box auditing approach relies on querying the model and analyzing its outputs against pre-defined fairness metrics (Birhane et al., 2024; de Vos et al., 2024). Current attempts to enhance fairness audits with tangible guarantees draw

inspiration from hypothesis testing (Si et al., 2021; Taskesen et al., 2021; DiCiccio et al., 2020; Cen & Alur, 2024; Cherian & Candès, 2024; Bénesse et al., 2024), online fairness auditing (Chugg et al., 2023; Maneriker et al., 2023), and formal methods for fairness certification (Albarghouthi et al., 2017; Ghosh et al., 2021; 2022; Borca-Tasciuc et al., 2022). Beyond statistical methods, the work of Yadav et al. explore the role of explanations in the auditing process (Yadav et al., 2022). Recent works also stress the importance of broadening the lens of algorithm auditing by incorporating user perspectives and sociotechnical factors (Lam et al., 2023; Deng et al., 2023). On another line of research, Confidential-PROFITT and FairProof propose to integrate cryptographic techniques in cooperation with the platforms, to ensure the faithfulness of platform responses during audits (Yadav et al., 2024; Shamsabadi et al., 2023; Waiwitlikhit et al., 2024); this is, however, more intrusive and technically restrictive, and thus awaits for adoption.

**Manipulating audits** Manipulating fairness audits is an active area of research. Auditors can be fooled by biased sampling when the decision maker is allowed to publish a labeled dataset as proof of model fairness (Fukuchi et al., 2020). Adversarial attacks on explanation methods, such as LIME and SHAP, can be employed to produce misleading interpretations of model behavior (Fokkema et al., 2023; Shamsabadi et al., 2022; Laberge et al., 2023; Slack et al., 2020; Anders et al., 2020; Aïvodji et al., 2019; Le Merrer & Trédan, 2020). Platforms can also modify the output of their models to create the appearance of fairness without addressing underlying biases (Yan & Zhang, 2022; Garcia Bourrée et al., 2023; Godinot et al., 2024).. However, the challenge of designing audits that are robust to advanced manipulation strategies remains open. The idea of using auditor prior knowledge that we formalize in this work has been implicitly studied in different contexts. Based on active learning techniques work has studied how auditors could leverage knowledge about the hypothesis class (Yan & Zhang, 2022; Godinot et al., 2024). In a more practical setting, Tan et al. studied using model distillation methods (Tan et al., 2018) to use prior about the ground truth and hypothesis class (Tan et al., 2018).

# 7. Conclusion and Discussion

We investigated, both theoretically and experimentally, the conditions under which an auditor can or cannot be manipulated when auditing with a prior. We introduced an empirical method for tuning the manipulation detection threshold to maximize the auditor's probability of detecting malicious platforms.

While our work offers regulators a framework for defending against audit manipulations, the path to accountability

extends much further. A significant gap remains between audit evaluations and the actual mitigation of identified issues (Raji et al., 2021; Mukobi, 2024). Moreover, one-time audits are inherently limited, as platforms can alter their models in harmful ways after the audit has concluded. Addressing these challenges in future work will require the development of continuous or adaptive auditing mechanisms, potentially incorporating auditor priors, to ensure sustained accountability and fairness.

## Impact Statement

This work provides both theoretical and empirical analyses of fairness audits in ML decision-making systems, with a focus on their vulnerability to strategic manipulations by platforms aiming to evade regulatory scrutiny. By demonstrating how auditors' access to prior knowledge can enhance the robustness of black-box audits, we offer actionable insights for mitigating potential audit-a manipulations. Our findings have important implications for policymakers, auditors, and ML practitioners, underscoring the urgent need for rigorous auditing frameworks resilient to adversarial behavior.

The societal impact of this work is twofold. On the positive side, strengthening the robustness of fairness audits promotes greater accountability for platforms deploying ML models in high-stakes domains such as finance and healthcare. By mapping the risk landscape of audit manipulation, our approach advances the development of more trustworthy ML systems. However, we also draw attention to the limitations of current audit practices, showing that over-reliance on public priors can be exploited by strategic actors.

## Acknowledgements

This work of Martijn de Vos, Milos Vujasinovic, Sayan Biswas, and Anne-Marie Kermarrec has been funded by the Swiss National Science Foundation, under the project "FRIDAY: Frugal, Privacy-Aware and Practical Decentralized Learning", SNSF proposal No. 10.001.796. Jade Garcia Bourrée, Augustin Godinot, Gilles Trédan and Erwan Le Merrer acknowledge the support of the French Agence Nationale de la Recherche (ANR), under grant ANR-24-CE23-7787 (project PACMAM).

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

*Table 1.* Notations

| | |
|---|---|
| $\mathcal{H}$ | HYPOTHESIS CLASS |
| $\mathcal{F}$ | SET OF FAIR MODELS |
| $\mathcal{H}_a$ | SET OF EXPECTABLE MODELS |
| $h_a$ | GROUND TRUTH |
| $\delta$ | DISTANCE BETWEEN THE GROUNDTRUTH AND THE SET OF EXPECTABLE MODEL |
| $h_p$ | ORIGINAL MODEL OF THE PLATFORM |
| $h_m$ | MANIPULATED MODEL OF THE PLATFORM |
| $\mathcal{X}$ | INPUT SPACE |
| $\mathcal{D}$ | DATA DISTRIBUTION |
| $X$ | SAMPLE FROM INPUT SPACE |
| $\mathcal{Y}$ | OUTPUT SPACE |
| $Y$ | SAMPLE FROM OUTPUT SPACE |
| $\mathcal{A}$ | PROTECTED FEATURE |
| $\mathcal{Z}$ | SAMPLE SPACE |
| $Z$ | SAMPLE |
| $n$ | DIMENSION OF $\mathcal{Z}$ |

# A. Proofs and additional theoretical results

As in (Buyl & Bie, 2022), let $\mathcal{Z} \triangleq \mathcal{X} \times \mathcal{A} \times \{0, 1\}$ denote the sample space, from which the auditor draws samples $Z \triangleq (X, A, Y)$. The auditor sample the binary predictions $\hat{Y} \in \{0, 1\}$ from a probabilistic classifier $h : \mathcal{X} \to [0, 1]$ that assigns a score $h(X)$ to the belief that a sample with features $X$ belongs to the positive class. It is assumed that $\mathcal{X} \subset \mathbb{R}^{d_\mathcal{X}}$ and $\mathcal{A} = \{[A_0, A_1]/A_0, A_1 \in \{0, 1\}\} = \{[1, 0], [0, 1]\}$(the one-hot encoding of the protected feature with two groups). We also assume that $\mathcal{H}_a$ is an open set of $\mathcal{Z}$.

We denote $\mathcal{F}$ the set of all score functions $f : \mathcal{X} \to \{0, 1\}$ that satisfy (PDP):

$$\mathcal{F} \triangleq \{f : \mathcal{X} \to \{0, 1\} : \mathbb{E}_Z [g(Z)f(X)] = 0_n\}$$

with $\forall k \in [2], g_k = \frac{A_k}{\mathbb{E}_Z[A_k]} - 1$, $0_n$ a vector of $n = d_\mathcal{F}$ zeros.

Assuming that the predictions $\hat{Y}|X$ are randomly sampled from a probabilistic classifier $h(X)$, then the traditional fairness notion of demographic parity (DP) is equivalent to PDP. But if $\hat{Y}$ is not sampled from $h(X)$ but instead decided by a threshold, DPD is a relaxation of the actual DP notion. That is to say, $\mathcal{F}$ is the set of all score functions that are fair regarding the demographic parity on $\mathcal{A}$.

As $\mathcal{F}$ is the kernel of the linear transformation $f : \mathbb{E}_Z [g(Z)f(X)]$, $\mathcal{F}$ is a hyperplane of $\mathcal{Z}$.

As $\mathcal{F}$ is a hyperplane of $\mathcal{Z}$, it is dense or closed in $\mathcal{Z}$.

## A.1. Cases where $\mathcal{F}$ is dense in $\mathcal{Z}$.

**Lemma A.1.** *If $\mathcal{F}$ is dense in $\mathcal{Z}$, the auditor has a probability to detects it as manipulated equals to zero.*

*Proof.* If $\mathcal{F}$ is dense in $\mathcal{Z}$ then for every function $f \in \mathcal{Z}$, every open neighborhood of $f$ intersects $\mathcal{F}$. In particular, it always exists a model $h_m \in \mathcal{F}$ that is in a neighborhood of $h_p$ and in $\mathcal{H}_a$. In that case, $h_m$ is fair and expectable, so the auditor has a probability to detects it as manipulated equals to zero. $\square$

This case is a pathological case where the platform can still appear fair and honest. For the next theoretical results, we are interested in the case where $\mathcal{F}$ is closed in $\mathcal{Z}$.

## A.2. Cases where $\mathcal{F}$ is closed in $\mathcal{Z}$.

If $\mathcal{F}$ is a hyperplan closed in $\mathcal{Z}$, it has an empty interior (*i.e.* $\partial\mathcal{F} = \emptyset$) as its codimension is 1. In the following, we can thus use $\mathcal{F}$ instead of $\partial\mathcal{F}$, as both are equals.

Similarly, we can define the normal vector to $\mathcal{F}$ which is actually the vector that is used for all the projections we use in this paper. In Equation (6), we defined $h_m^* = \text{proj}_{\mathcal{F}}(h_p)$ (i.e. $h_m^*$ is the orthographic projection of the expectable model $h_p$ in the set of fair models $\mathcal{F}$).

Having an hyperplan lead to the natural definition of (hyper)cylinder, that we use in the following theorem.

**Definition A.2.** A right cylinder $C(H, B)$ is the set of all points whose orthographic projection on a hyperplane $H$ lies in a set $B$ with $B$ a subset of the boundary of $H$. $B$ is called the base of the cylinder.

**Theorem A.3.** *The probability $P_{uf}$ that the auditor correctly detects a malicious platform trying to be fair is* $P(\mathcal{H}_a \backslash C(\mathcal{F}, \mathcal{H}_a \cap \partial \mathcal{F})|\mathcal{H}_a)$.

*Proof.* The auditor correctly detects a malicious platform trying to be fair if and only if the manipulated model is fair but not expectable. The manipulated model is fair but not expectable if and only if the orthographic projection $h_m^*$ of $h_p$ in $\mathcal{F}$ is not in $\mathcal{H}_a \cap \partial \mathcal{F}$. Thus, the manipulated model is fair but not expectable if and only if $h_p \notin C(\mathcal{F}, \mathcal{H}_a \cap \partial \mathcal{F})$ (following Definition A.2). As by assumption $h_p \in \mathcal{H}_a$ (Equation (4)), it means that $h_p \in \mathcal{H}_a \backslash C(\mathcal{F}, \mathcal{H}_a \cap \partial \mathcal{F})$. The auditor correctly detects a malicious platform trying to be fair with probability $P(\mathcal{H}_a \backslash C(\mathcal{F}, \mathcal{H}_a \cap \partial \mathcal{F})|\mathcal{H}_a)$. $\qquad \square$

Theorem 4.3 is a special case of Theorem A.3 with additional assumption. We now prove the main Theorem Theorem 4.3.

**Theorem 4.3** (Prior-Uniform detection rate)**.** *Under the dataset prior of Definition 4.1 with $L$ defined as the $\ell_2$ norm, and the* uninformative prior *assumption, the probability that the auditor correctly detects a malicious platform trying to be fair is*

$$1 - \frac{1}{W_n} \left( \int_0^{\arccos(\delta/\tau)} \sin^n(\theta) d\theta - \frac{\delta}{\tau} \left( 1 - \frac{\delta^2}{\tau^2} \right)^{(n-1)/2} \right).$$

*with $\delta = d(h_a, \mathcal{F})$, the distance of $h_a$ to $\mathcal{F}$ and $W_n$ is the $n$-term of Wallis' integrals.*

*Proof.* As established in Theorem A.3, $P_{uf} = P(\mathcal{H}_a \backslash C(\mathcal{F}, \mathcal{H}_a \cap \partial \mathcal{F})|\mathcal{H}_a)$.

The probability $P(\mathcal{H}_a \backslash C(\mathcal{F}, \mathcal{H}_a \cap \partial \mathcal{F})|\mathcal{H}_a)$ is the probability to be in the ball $\mathcal{H}_a$ without the probability to be in the intersection between the ball $\mathcal{H}_a$ and the cylinder $C(\mathcal{F}, \mathcal{H}_a \cap \partial \mathcal{F})$. In the following, we denote $V_n^{\#}(\tau, \delta)$ this quantity.

As $\mathcal{H}_a$ is a ball, its volume is:

$$V_n^{ball}(\tau) = \frac{\pi^{n/2} \tau^n}{\Gamma(\frac{n+2}{2})}$$

with $\Gamma(z) = \int_0^{\infty} t^{z-1} e^{-t} dt$ (NIST, 2013).

The volume of the intersection between the cylinder and the ball is the sum of the three following volumes:

- the solid cylinder with height between $-\delta$ and $\delta$

- the spherical cap of $\mathcal{H}_a$ that is *above* the previous cylinder (i.e. the part of $\mathcal{H}_a$ with height between $\delta$ and $\tau$)

- the spherical cap of $\mathcal{H}_a$ that is *bellow* the previous cylinder (i.e. the part of $\mathcal{H}_a$ with height between $-\delta$ and $-\tau$)

According to (Li, 2010), the volume of each spherical cap is

$$V_n^{cap}(\tau, \delta) = \frac{\pi^{(n-1)/2} \tau^n}{\Gamma(\frac{n+1}{2})} \int_0^{\arccos(\delta/\tau)} \sin^n(\theta) d\theta$$

And the volume of the cylinder of height $2\delta$ is

$$V_n^{cylinder}(\tau, \delta) = 2\delta V_{n-1}^{ball}(\sqrt{\tau^2 - \delta^2})$$

Thus,

$$V_n^{\#}(\tau, \delta) = V_n^{ball}(\tau) - 2V_n^{cap}(\tau, \delta) - V_n^{cylinder}(\tau, \delta)$$

$$= \frac{\pi^{n/2}\tau^n}{\Gamma(\frac{n+2}{2})} - 2\frac{\pi^{(n-1)/2}\tau^n}{\Gamma(\frac{n+1}{2})} \int_0^{\arccos(\delta/\tau)} \sin^n(\theta)d\theta - 2\delta\frac{\pi^{(n-1)/2}(\sqrt{\tau^2-\delta^2})^{n-1}}{\Gamma(\frac{n+1}{2})}$$

According to Theorem A.3, the probability that the auditor correctly detects a malicious platform trying to be fair is $P(\mathcal{H}_a\backslash C(\mathcal{F}, \mathcal{H}_a \cap \partial\mathcal{F})|\mathcal{H}_a)$. That is to say, it is the ratio of $V_n^\#(\tau, \delta)$ over $V_n^{ball}(\tau)$:

$$\begin{aligned}
P_{uf} &= P(\mathcal{H}_a\backslash C(\mathcal{F}, \mathcal{H}_a \cap \partial\mathcal{F})|\mathcal{H}_a)\\
&= \frac{V_n^\#(\tau, \delta)}{V_n^{ball}(\tau)}\\
&= 1 - 2\frac{\Gamma(\frac{n+2}{2})}{\Gamma(\frac{n+1}{2})}\frac{\pi^{(n-1)/2}}{\pi^{n/2}}\int_0^{\arccos(\delta/\tau)}\sin^n(\theta)d\theta - 2\delta\frac{(\tau^2-\delta^2)^{(n-1)/2}}{\tau^n}\frac{\Gamma(\frac{n+2}{2})}{\Gamma(\frac{n+1}{2})}\frac{\pi^{(n-1)/2}}{\pi^{n/2}}\\
&= 1 - \frac{2}{\sqrt{\pi}}\frac{\Gamma(\frac{n+2}{2})}{\Gamma(\frac{n+1}{2})}\int_0^{\arccos(\delta/\tau)}\sin^n(\theta)d\theta - \frac{2\delta}{\sqrt{\pi}}\frac{(\tau^2-\delta^2)^{(n-1)/2}}{\tau^n}\frac{\Gamma(\frac{n+2}{2})}{\Gamma(\frac{n+1}{2})}\\
&= 1 - \frac{2}{\sqrt{\pi}}\frac{\Gamma(\frac{n+2}{2})}{\Gamma(\frac{n+1}{2})}\left(\int_0^{\arccos(\delta/\tau)}\sin^n(\theta)d\theta - \delta\frac{(\tau^2-\delta^2)^{(n-1)/2}}{\tau^n}\right)
\end{aligned}$$

The function $\Gamma$ can be written with Wallis' integrals as: $W_n = \frac{\sqrt{\pi}}{2}\frac{\Gamma(\frac{n+1}{2})}{\Gamma(\frac{n+2}{2})}$ with $\forall n, W_n = \int_0^{\pi/2}\sin^n(\theta)d\theta$.

In the other hand,

$$\begin{aligned}
\delta\frac{(\tau^2-\delta^2)^{(n-1)/2}}{\tau^n} &= \frac{\delta}{\tau}\frac{(\tau^2-\delta^2)^{(n-1)/2}}{\tau^{n-1}}\\
&= \frac{\delta}{\tau}\left(\frac{\tau^2-\delta^2}{\tau^2}\right)^{(n-1)/2}\\
&= \frac{\delta}{\tau}\left(1 - \frac{\delta^2}{\tau^2}\right)^{(n-1)/2}
\end{aligned}$$

Thus, $P_{uf} = 1 - \frac{1}{W_n}\left(\int_0^{\arccos(\delta/\tau)}\sin^n(\theta)d\theta - \frac{\delta}{\tau}\left(1 - \frac{\delta^2}{\tau^2}\right)^{(n-1)/2}\right)$. $\qquad\square$

Before dealing with this complete expression, we propose some particular cases that are easily interpretable.

**Corollary A.4.** *If $\mathcal{H}_a$ is a ball centered in the ground-truth $h_a$ that is tangent to $\mathcal{F}$, then the auditor has a probability one to correctly detect a malicious platform trying to be fair.*

$$\mathcal{H}_a = B(h_a, \tau) \wedge \tau = \delta \implies P_{uf} = 1.$$

*with $\delta = d(h_a, \mathcal{F})$, the distance of $h_a$ to $\mathcal{F}$.*

*Proof.* If $\mathcal{H}_a$ is tangent to $\mathcal{F}$ then $\delta = \tau$. Thus, $\arccos(\delta/\tau) = \arccos(1) = 0$ and $\int_0^{\arccos(\delta/\tau)}\sin^n(\theta)d\theta = 0$.

Thanks to the formula of Theorem 4.3 with $\delta/\tau = 1$, $P_{uf} = 1 - \frac{1}{W_n}(0 - 0) = 1$.

If $\mathcal{H}_a$ is tangent to $\mathcal{F}$, $P_{uf} = 1$. $\qquad\square$

This corollary means that by reducing the threshold $\tau$ to the minimal value ($\delta$), the auditor is sure to detect any manipulation of the platform.

**Corollary A.5.** *If $\mathcal{H}_a$ is a ball centered in the ground-truth $h_a$ that is fair, then the auditor has a probability zero to correctly detect a malicious platform trying to be fair.*

$$\mathcal{H}_a = B(h_a, \tau) \wedge h_a \in \partial\mathcal{F} \implies P_{uf} = 0.$$

*Proof.* If $h_a \in \partial\mathcal{F}$ then $\delta = 0$ and $\arccos(\delta/\tau) = \arccos(0) = \pi/2$ in the formula of Theorem 4.3. Thus, $P_{uf} = 1 - \frac{1}{W_n}(W_n - 0) = 0$.

$\square$

This last case is the case where the $h_a$ of the auditor is fair. Intuitively, if $h_a$ is fair, half of the model that the platform can construct are naturally fair and the other half are naturally unfair. Thus, it is very easy to change from an unfair model to a fair model without changing too much the honest model. Thus, detecting such manipulation is very hard for the auditor.

Now, we study the general expression of $P_{uf}$ in Theorem 4.3. In particular, we study a lower bound of $P_{uf}$ to study when the probability is strictly positive.

**Corollary 4.4** (Detection rate lower bound). *If $n$ is even,*

$$\frac{1}{W_n}\frac{\delta}{\tau}\left(1 - \frac{\delta^2}{\tau^2}\right)^{(n-1)/2} \leq P_{uf} \leq 1.$$

*Proof.* $P_{uf} = 1 - \frac{1}{W_n}\left(\int_0^{\arccos(\delta/\tau)} \sin^n(\theta)d\theta - \frac{\delta}{\tau}\left(1 - \frac{\delta^2}{\tau^2}\right)^{(n-1)/2}\right)$

$W_n = \int_0^{\arccos(\delta/\tau)} \sin^n(\theta)d\theta + \int_{\arccos(\delta/\tau)}^{\pi/2} \sin^n(\theta)d\theta$

So, $\frac{1}{W_n}\int_0^{\arccos(\delta/\tau)} \sin^n(\theta)d\theta \leq 1$ (with n even).

And $P_{uf} \geq \frac{1}{W_n}\frac{\delta}{\tau}\left(1 - \frac{\delta^2}{\tau^2}\right)^{(n-1)/2}$

$\square$

**Lemma A.6.** *The lower bound according to $\delta/\tau$ has two extremums that are for $\delta/\tau = 1$ or $\delta/\tau = \gamma$ with $\gamma = \frac{\sqrt{n+3} - \sqrt{n-1}}{2}$.*

**Remark.** Note that $\gamma$ only depend on the dimension $n$ and leads to $0$ when $n$ leads to infinity.

*Proof.* We define $f_n$ (the lower bound) s.t.

$$f_n(\delta, \tau) = \frac{\delta}{W_n \tau}\left(1 - \frac{\delta^2}{\tau^2}\right)^{(n-1)/2}$$

Change of variable $x = \frac{\delta}{\tau}$, $f(x) = \frac{x}{W_n}(1 - x^2)^{(n-1)/2}$.

We are interested in cases where $\tau > \delta$, *i.e.* $0 < x < 1$.

Moreover, $f$ has an extremum iff $f' = 0$ somewhere in $[0, 1]$.

$$\forall x \in [0, 1], W_n f'(x) = (1 - x^2)^{(n-1)/2} - (n-1)x^2(1 - x^2)^{(n-3)/2}$$
$$= (1 - x^2)^{(n-3)/2}(x^2 + \sqrt{n-1}x - 1)(x^2 - \sqrt{n-1}x - 1)$$

*i.e.* $f'(x) = 0$ for the following elements:

- $x = -1 < 0$
- $x = 1$

- $\frac{-\sqrt{n-1}-\sqrt{n+3}}{2} < 0$

- $\frac{-\sqrt{n-1}+\sqrt{n+3}}{2} \in [0, 1]$

- $\frac{\sqrt{n-1}-\sqrt{n+3}}{2} < 0$

- $\frac{\sqrt{n-1}+\sqrt{n+3}}{2} > 1$ (if $n \geq 2$)

So $f$ has two local extremums in $[0, 1]$, one for 1 and one for $\gamma = \frac{\sqrt{n+3}-\sqrt{n-1}}{2}$. $\qquad\square$

