# OpenReview forum: "Robust ML Auditing using Prior Knowledge"
_ICML.cc/2025/Conference — ICML 2025 spotlightposter_

### Official Review · Reviewer_Q6QF · 2025-02-23

**Overall Recommendation:** 5

**Summary:**

Audits have been historically impactful in AI and they are increasingly becoming a common part of proposals for regulating it. however, it could be possible for developers to game audits so that the model seems to behave much better on the evaluation than real-world cases. This paper discusses this problem, raises the alarm about gaming publicly available evaluations, and formally describe the conditions by which audits can be manipulated. Key to this paper is introducing the concept of an auditor prior which describes what companies think about how their systems are going to be evaluated.

**Claims And Evidence:**

I would've been happy with this kind of paper as a position paper that didn't do any experiments. The fact of this paper did any experiments at all reflects a relatively thorough approach to the type of work that they're doing here.

**Essential References Not Discussed:**

- https://arxiv.org/abs/2408.02565
- https://dl.acm.org/doi/abs/10.1145/3630106.3659037
- https://arxiv.org/abs/2111.15366

**Experimental Designs Or Analyses:**

The authors focus on evals for fairness. This seems fine to me, but I'm not sure why they don't take us slightly more general approach and make this paper about evaluations in general of fairness or other properties.

I like how the authors did not just focus on instances in which developers know the actual evaluations because they are based on publicly available resources. Focusing on the more general case is great because it's very realistic. It's a very real problem that developers can predict evaluations, even without knowing what they are exactly.

**Methods And Evaluation Criteria:**

I like the experiments. I think it's smart how they quantified concealable unfairness.

**Other Comments Or Suggestions:**

I would recommend trying to move figure one up in the paper to be on page one or two.

**Other Strengths And Weaknesses:**

Overall, I really like this paper. I think it should be accepted. My one hangup about it is that I think it would be somewhat better if it were possible to make this paper about evaluations in general rather than just evaluations of fairness. One other small thing is that the title kind of confuses me since the prior knowledge is the problem rather than the thing that makes an auto robust.

**Questions For Authors:**

None to add. Please respond to points above.

**Relation To Broader Scientific Literature:**

Overall, I think this is a great paper. As a reviewer, I would be willing to go to bat for it. I found it personally clarifying, well executed, and of clear value to the literature. I can easily see myself citing this paper.

**Theoretical Claims:**

I have not checked all of the details and verified that the math works out myself. But the theory here is in service of the experiments which are done well. One could also argue that the theory here has the primary purpose of being illustrative since it's based on assumptions. There will be fundamental limitations in how easily real world phenomena can be captured with simple models like this. I think this is a no way of weakness of the paper -- this is me just saying that the theory's role in their arguments is illustrative rather thanbeing the core.

---

> ### Author Rebuttal · Authors · 2025-03-31
>
> Dear Reviewer Q6QF,
>
> Thank you very much for taking the time to read and review our paper. We are delighted that you found such value in our work and agree that it is a very real problem that developers can predict evaluations.
>
> **How about more general evaluations?**
> In this paper, we chose to focus on demographic parity because it can be very easily manipulated by a malicious platform (answering uniformly at random gives perfect demographic parity). We are encouraged by the results in this difficult setting. Therefore, testing other evaluation metrics in the manipulation-proof framework (and comparing the brittleness of the metrics we currently use) is, in fact, our end goal.
> About your question on the title and prior knowledge, we would like to refer you to our answer to Reviewer g7Ct (Security Game paragraph) for more clarification.
>
> **Discussion on additional references**
> We thank the reviewer for pointing to the work of [Mukobi](https://arxiv.org/abs/2408.02565), we were not aware of it. The other papers discuss different shortcomings of current ML evaluation methods (in terms of safety evaluations for [Casper et al.](https://dl.acm.org/doi/abs/10.1145/3630106.3659037) and capacity evaluations for [Raji et al.](https://arxiv.org/abs/2111.15366)).
> While we did include a reference to another study on how these evaluations are conducted ([Birhane et al.](https://ieeexplore.ieee.org/document/10516659)), we agree that this discussion on the gap between evaluations and mitigations deserves more attention. Thus, we will add the references you mentioned along with the discussion to the Related Works section.
>
> We thank you again for your positive evaluation of our work and hope that we have addressed your concerns. Should you have additional questions or remarks, we would be happy to answer them in the final discussion phase.

---

### Official Review · Reviewer_Ddwq · 2025-03-05

**Overall Recommendation:** 3

**Summary:**

This paper addresses a significant challenge in machine learning fairness auditing: the risk of manipulation (fairwashing) during audits. The authors introduce an approach to make audits more robust by incorporating the auditor's prior knowledge about the task. Through theoretical analysis and experiments, they establish conditions under which auditors can prevent audit manipulations and quantify the maximum unfairness a platform can conceal. The work formalizes how auditors can leverage their private knowledge through labeled datasets to detect platforms that artificially modify their models during audits to appear fair.

**Claims And Evidence:**

The key claims and supporting evidence include:

1. Public priors are insufficient. The authors theoretically prove that if the platform knows the auditor's prior, it can always manipulate the audit to appear fair and honest.
2. Private dataset priors improve detection. The paper demonstrates both theoretically and experimentally that auditors can prevent manipulations by maintaining a private labeled dataset.
3. Quantifiable manipulation protection. The authors derive a mathematical framework to calculate the probability of detecting manipulations based on the auditor's dataset and its relation to the fair model space.

**Essential References Not Discussed:**

The paper covers the most relevant literature but could benefit from discussing:

1. Additional work on formal verification of fairness guarantees beyond what's mentioned
2. Recent advances in interactive auditing approaches that might complement their dataset prior approach

**Experimental Designs Or Analyses:**

The experimental design is thorough but conventional:

1. The datasets used (CelebA and ACSEmployment) are standard benchmarks in fairness literature
2. The model types evaluated (GBDT, Logistic Regression, LeNet) are commonly used in comparable studies
3. The manipulation strategies tested are adaptations of existing fairness repair methods

The experiments align with expectations based on the theoretical analysis but do not provide any surprising or novel insights into fairness auditing. The models implemented in this paper are relatively early, classic works; incorporating more innovative or up-to-date approaches would be preferable.

**Methods And Evaluation Criteria:**

The paper's methodological approach is competent but lacks novelty:

1. The probability bounds for detecting manipulation are well-derived.
2. The experiments on tabular data (ACSEmployment) and image data (CelebA) are thorough. I question whether the amount of data in the datasets is sufficient. And whether using only the "concealable unfairness" metric throughout the paper is adequate.
3. Practical implementation: The four fairness repair methods adapted as manipulation strategies are all pre-existing techniques in the literature. It would be more convincing if there were newer methods.

**Other Comments Or Suggestions:**

None

**Other Strengths And Weaknesses:**

Strengths:

1. The paper addresses a practical and important problem in ML ethics and regulation.
2. The theoretical analysis is rigorous and provides useful bounds.

Weaknesses:

1. The paper assumes the auditor can collect a representative labeled dataset, which may be challenging in practice.
2. The theoretical analysis primarily focuses on demographic parity; extensions to other fairness metrics would strengthen the contribution.
3. The discussion of practical implementation in regulatory contexts could be expanded.

Minor:
1. Line 78: "where where"
2. Line 181: "a a"

While technically sound, the paper represents an incremental advance rather than a substantial novel contribution to fairness auditing research. Leveraging private knowledge to improve audit robustness is a natural extension of existing work rather than a groundbreaking new direction.

**Questions For Authors:**

None

**Relation To Broader Scientific Literature:**

The paper effectively positions itself within the fairness auditing literature, acknowledging prior work on fairwashing through biased sampling, explanation manipulation, and black-box interactions. The authors clearly articulate their novel contribution: introducing a theoretical framework for incorporating private prior knowledge to prevent manipulations.

**Theoretical Claims:**

The theoretical framework is mathematically sound:

1. The formalization of the auditor's prior knowledge (Definitions 3.1 and 4.1) builds directly on established notions in the literature.
2. The relationship between prior knowledge and detection probability (Theorem 4.3) is predictably based on the basic principles of hypothesis testing and geometric interpretations of fairness.
3. The bounds on detection probability (Corollary 4.4) are well-derived.

The description related to theoretical claims is abstract, making it difficult for readers to understand the content of the formulas.

---

> ### Author Rebuttal · Authors · 2025-03-31
>
> Dear Reviewer Ddwq,
>
> Thank you for taking the time to read and review our paper. We are delighted that you found our work to be scientifically sound with a *thorough experimental design* and a *mathematically sound theoretical framework* that also *effectively positions itself within the fairness auditing literature*.
>
> **To be or not to be a groundbreaking new direction**
> We believe that our work is particularly relevant and of interest to the ICML community because:
> - Existing manipulation-proof auditing/fairwashing literature all implicitly use this notion of auditor prior. We formalize it, derive general theorems (Section 3), and introduce a new (and *natural*) type of prior (Section 4). See for example [Yan et al.](https://proceedings.mlr.press/v162/yan22c/yan22c.pdf), [Aivodji et al.](https://proceedings.mlr.press/v97/aivodji19a.html), [Shamsabadi et al.](https://proceedings.neurips.cc/paper_files/paper/2022/hash/5b84864ff8474fd742c66f219b2eaac1-Abstract-Conference.html) or [Yadav](https://proceedings.mlr.press/v235/yadav24a.html).
> - The ML security community is well versed in this type of security game formulation but they try to solve them with cryptographic primitives. We believe that the theoretical ML community can also bring nice audit guarantees. Hence, our auditor prior formulation opens up an avenue to explore better and potentially more complex notions of auditor priors.
>
> **Standard fairness repair methods**
> As manipulation strategies, we adapted fairness methods based on their applicability to our setting (binary classification, access to sensitive attributes, and compute overhead) and choose the best performing ones. If you have a specific more innovative or up-to-date method, we would be glad to try it in our experiments before the end of the discussion phase.
>
> **Further discussion on formal verification of fairness**
> Formal verification of fairness methods (see Section 6) adapts classical verification frameworks (SSAT, bound propagation...) to fairness auditing. To the best of our knowledge, all require a white box access to the model. Thus, while they are very useful for model providers, existing formal fairness verification methods are not applicable in our setting where the auditor only has black box access.
>
> We have corrected the typos you pointed out in the manuscript. We thank you again for your comments on our work and hope that our contributions are now clearer. Should you have additional questions or remarks, we would be happy to answer them in the final discussion phase.

---

### Official Review · Reviewer_g7Ct · 2025-03-14

**Overall Recommendation:** 2

**Summary:**

The paper studies the problem of robust fairness auditing when the platform can manipulate the model during auditing. To address this problem, they propose to allow the auditor to have access to a set of labeled examples that are close to the prediction of the model before auditing. During the auditing, the auditor performs two tests: whether the audited model is manipulated (by checking whether the labeled examples are still close to the prediction of the model during auditing) and whether the audited model is fair. Theoretical analysis derives upper and lower bounds for the successful detection probability, under simplified assumptions of the closeness. The experimental analysis uses existing fairness repair methods as model manipulation strategies and quantifies the dynamics of conceivable unfairness as the auditing budget grows.

**Claims And Evidence:**

1. Theorem 4.3. analyzes the detection success probability, yet it does not specify the randomness over which the probability is computed. Theorem 4.3 also requires some assumptions on the closeness and model distribution, yet these assumptions are not covered in the statement.
2. Experiments in Sections 5.3 and 5.4 did not report how the closeness threshold $\tau$ is chosen for detecting model manipulation. This is quite important as the results (Figure 4) could drastically change under a different threshold. The threshold choice is also highly dependent on the task and data distribution, which requires detailed justifications.

**Essential References Not Discussed:**

NA

**Experimental Designs Or Analyses:**

See Claims And Evidence and Methods And Evaluation Criteria.

**Methods And Evaluation Criteria:**

The proposed method, theoretical assumptions, as well as evaluation criteria are overly simplified.
1. The method is essentially collecting predictions of the model before auditing, and using them to detect whether a model is manipulated during auditing. To tolerate the realistic settings where the auditor could not have precisely accurate predictions by the model before auditing, the authors make assumptions about how close the collected data are to the actual model predictions. Yet this assumption is highly application-specific. In the current form, the authors simply assume a fixed threshold for l2 norm or accuracy closeness, which doesn't reflect practical auditing applications.
2. The assumption of the model owner not knowing the data prior is somewhat problematic, and resembles the line of thinking of "security by obscurity".

**Other Comments Or Suggestions:**

I enjoyed reading the paper, and appreciate the importance of the discussed problem. However, I believe the approach taken in this paper requires more justifications in many assumptions: the closeness metric/threshold, knowledge of the platform about the data prior...See Claims And Evidence and Methods And Evaluation Criteria. Additionally, a discussion about (why not using) alternative solutions could be helpful, e.g., asking the platform to sign each of its predictions, thus providing proof that all predictions are produced by the audited model.

Minor typo: Definition 4.2 $h^*_m\in \mathcal{H}_a$ should be $h^*_m\notin \mathcal{H}_a$

**Other Strengths And Weaknesses:**

See Claims And Evidence and Methods And Evaluation Criteria.

**Questions For Authors:**

See Other Comments Or Suggestions.

**Relation To Broader Scientific Literature:**

The paper revisited the data prior for robust fairness auditing.

**Theoretical Claims:**

Proofs look correct yet the statement requires more clarifications. See Claims And Evidence.

---

> ### Author Rebuttal · Authors · 2025-03-31
>
> Dear Reviewer g7Ct,
>
> Thank you very much for taking the time to read and review our paper. We are delighted that you *enjoyed reading the paper* and that, as reviewer Ddwq and Q6QF, you *appreciate the importance of the discussed problem*.
>
> We will answer your points on the security game formulation, threshold value, and alternative solutions, but first, we would like to address what we believe is a misunderstanding. You mention that *The method is essentially collecting predictions of the model before auditing*, whereas the method we instantiate in Section 4 is about collecting *ground truth* labels before the audit.
>
> **Security game**
> To the best of our understanding, a "Security by obscurity" approach would try to hide the verification protocol hoping that the platform could thus not manipulate it. In our setting, the audit protocol is public: the platform knows what metric is measured and knows that the auditor will use a prior (e.g., ground truth labels in Section 4). What is private is "the key": the exact realization of the ground truth label, and thus the exact models in $\mathcal{H}_a$. Similar audit protocols can be found in other domains such as accounting. The protocol is public (the company knows that the auditor will look at their records but not all because of resource constraints) but the exact transactions that are examined by the auditor are not known to the auditee beforehand.
>
> **Theorem 4.3 assumptions**
> Since the auditor has no prior bias or belief on the model used by the platform, following the notion of *uninformative prior*, we assume that the auditor considers all models in $\mathcal{H}_a$ to be equiprobable. This assumption and the closeness axiom (i.e., the auditor prior is a *good* prior) are justified in lines 174-186 (1st column) and lines 245-248 (2nd column). We will update the Theorem 4.3 statement to include and better justify the uniform model distribution assumption.
>
> **Decision threshold $\tau$ value**
> We agree that the *threshold choice is also highly dependent on the task and data distribution*. It is important in practice for the auditors to understand how to choose $\tau$ and this choice can be very much context- and application-specific. We describe exactly how it is chosen in practice in Section 5.4, lines 374-378. We agree that the paper would benefit from an earlier discussion of this setup. We will add it next to Definition 4.1.
>
> **Alternative solutions**
> Finding ways to *ask the platform to sign each of its predictions* is indeed a very active research area for manipulation proof auditing. In the related works we mention some early efforts in using cryptographic primitives for auditing (Yadav et al., 2024; Shamsabadi et al., 2023; Waiwitlikhit et al., 2024). However, beyond the extremely high computational and infrastructure cost of signing model predictions, our paper provides an alternative with a more learning theoretic view on the manipulation-proof auditing problem by deriving the audit guarantees from the auditor's expertise (i.e., prior).
>
> We have corrected the typos you pointed out in the manuscript. We thank you again for your comments on our work and hope that we have satisfactorily addressed your main concerns. Should you have additional questions or remarks, we would be happy to answer them in the final discussion phase.

---

> > ### Comment · Reviewer_g7Ct · 2025-04-04
> >
> > The analogy to data prior to key distribution in cryptography is intriguing. If I understand correctly, conceivable fairness now aligns with key strength—serving more as a passive measure of audit uncertainty or robustness, rather than something actively controllable. (Indeed, in Figure 4, the best manipulation could achieve non-negligible conceivable fairness under ACSEmployment dataset even under large auditing budget.) If so, I'm not sure how useful the robust audit is. It would be helpful if the authors could discuss potential ways an auditor might reduce conceivable fairness, such as modifying the model space.
> >
> > All other concerns have been addressed—thank you.

---

> > > ### Author Response · Authors · 2025-04-07
> > >
> > > Dear Reviewer g7Ct,
> > >
> > > Thank you for your comments, we are glad your previous points have been adressed.
> > >
> > > Indeed, we introduced the notion of concealable unfairness as a measure of the robustness of an audit strategy. In practice, concealable unfairness is influenced by the data distribution, the platform's manipulation strategy, and the auditing strategy.
> > >
> > > As you pointed out, in Figure 4, there are a few cases where the concealable unfairness (i.e., how much unfairness the platform was able to hide by manipulating its answers) is still non-negligible even at high audit budgets. It is important to note that Figure 4 presents a worst-case analysis: among all the models we simulated, we picked the ones for which the manipulation was most effective.
> > >
> > > Here are two potential ways an auditor might reduce this concealable unfairness.
> > > 1. Tune the detection threshold. Figure 4 used a conservative strategy to set the manipulation detection threshold $\tau$. Figure 3 shows that if the auditor is able to tune the threshold well enough, the concealable unfairness can be brought down to 0. Of course, if the auditor does not even trust their labels, a label-based robust audit does not make sense anymore. Thus, understanding the connections between the uncertainty of the task and the achievable audit guarantees would be a very nice follow-up work.
> > > 2. Improve the auditor prior. In Section 4, we instantiate the robust audits framework with one strategy: labeled audit dataset. Now assume that the auditor is able to train models similar to the platform's model (i.e., the auditor gains knowledge about the hypothesis class of the platform). In this case, instead of considering that the distribution of models inside $\mathcal{H}_a$ is uniform (by lack of more knowledge), the auditor can now have a better estimate of the model distribution. Finding efficient ways to incorporate this hypothesis class knowledge into the auditing procedure would also be an interesting avenue for future work.
> > >
> > > We shall add this discussion in the last part of the paper. Again, should you have additional questions or remarks, we would be happy to continue the discussion.

---

### Decision · Program_Chairs · 2025-05-01

**Decision:**

Accept (spotlight poster)

**Comment:**

This paper received diverse scores, ranging from "Weak Reject" to "Strong Accept," indicating a significant disagreement among reviewers.
A central discussion point was the extent to which practical implications should be required.
Some reviewers expressed concerns that the experimental setup and theoretical framework were too simplistic and might lack relevance to real-world problems.
Other reviewers claimed that the paper makes a valuable contribution to the field of AI auditing, with strong practical relevance.
All reviewers agreed that the authors should provide a more detailed discussion of the limitations of their theory and experiment.

I, as AC, think that the paper has the potential to open a new subfield in auditing.
Thus, detailed real-world practicality will be explored in future work.
We might not need to demand practical, real-world level insights in this initial paper.